# Underutilization Versus Nutritional-Nutraceutical Potential of the *Amaranthus* Food Plant: A Mini-Review

Olusanya N. Ruth [1,*], Kolanisi Unathi [1,2], Ngobese Nomali [3] and Mayashree Chinsamy [4]

1 Disipline of Food Security, School of Agricultural, Earth and Environmental Science University of KwaZulu-Natal, Scottsville, Pietermaritzburg 3209, South Africa; kolanisiU@unizulu.ac.za
2 Department of Consumer Science, University of Zululand, 24 Main Road, KwaDlangezwa, Uthungulu 3886, South Africa
3 Department of Botany and Plant Biotectechnology, University of Johannesburg, Auckland Park, Johannesburg 2092, South Africa; ngobesen@uj.ac.za
4 DST-NRF-Center, Indigenous Knowledge System, University of KwaZulu-Natal, Westville 3629, South Africa; Chinsamym@ukzn.ac.za
* Correspondence: 215081315@stu.ukzn.ac.za

**Abstract:** *Amaranthus* is a C4 plant tolerant to drought, and plant diseases and a suitable option for climate change. This plant could form part of every region's cultural heritage and can be transferred to the next generation. Moreover, *Amaranthus* is a multipurpose plant that has been identified as a traditional edible vegetable endowed with nutritional value, besides its fodder, medicinal, nutraceutical, industrial, and ornamental potentials. In recent decade Amaranthus has received increased research interest. Despite its endowment, there is a dearth of awareness of its numerous potential benefits hence, it is being underutilized. Suitable cultivation systems, innovative processing, and value-adding techniques to promote its utilization are scarce. However, a food-based approach has been suggested as a sustainable measure that tackles food-related problem, especially in harsh weather. Thus, in this review, a literature search for updated progress and potential uses of *Amaranthus* from online databases of peer-reviewed articles and books was conducted. In addition, the nomenclature, nutritional, and nutraceutical value, was reviewed. The species of focus highlighted in the review include, *A. blitum*, *A. caudatus*, *A. cruentus*, *A. dubius*, *A. hypochondriacus*, *A. spinosus*, *A. thunbergii*, *A. tricolor*, and *A. viridis*.

**Keywords:** *Amaranthus*; nutraceutical; nutrition security; potentials; traditional uses; underutilized vegetables

## 1. Introduction

There is great pressure on the agricultural sector to produce 70% of food that would feed the 40% projected population increase of nine billion people worldwide by the year 2050 [1]. Achieving such projections is a concern in research and, it calls for constant exploration of underutilized edible crops to provide food security for the world population. Additionally, the burden of hidden hunger has been linked with malnutrition, a food-related challenge. Infant, young children, young women of the age-bearing group in the middle- and low-income countries are at risk of the devastating effects, that impair vision, intellect, and retarded development as well as inflicting morbidity that limits livelihood of persons, especially the smallholder farmers in rural areas [1,2]. Interestingly, varieties of food plants that are drought tolerant like *Amaranthus* have been reported to have massive potential of curbing food-related problems, like micro-nutrient deficiencies challenge. Their contribution to food and nutrition security and consequential benefits that aimed at the improving wellbeing and livelihood is worth exploring. It is reported that about 30,000 plant species around the world are edible, with only 7000 that have been utilized as food [3]. Interestingly, many of these underutilized food plants grow naturally in many fallow landscapes, but people harvesting them are the most disadvantaged in the rural

community of many developing countries. Since the *Amaranthus* plant grows commonly, the indebt awareness of its potential and uses are limited. Little wonder most of the food crops, including *Amaranthus*, are underutilized. *Amaranthus* is a traditional crop with its origin traced to America, where it has been cultivated between 6000–8000 years ago, but currently, *Amaranthus* is cultivated across the globe [4]. Therefore, *Amaranthus* has been rediscovered among the few edible crops with a multipurpose potential that can be worth exploring. This is because of its diversified usefulness in industry. It has provided aesthetic value, with edible grains, leafy vegetables, fodder, and dietary sources compared to the most predominant staple crops [2,4]. Notwithstanding the geographical location of *Amaranthus* spp. It is recognized as a plant that grows within a short period [4]. However, there is a challenge in recognizing and classifying the species of *Amaranthus,* this is due to the critical morphological similarities, phenotypic variability, and cross-breeding that causes nomenclature disorders [2,5,6]. However, about 60–75 species of *Amaranthus* exist across the globe, 10 of which are dioecious (implying that, the male flowers are on one plant and the female flowers are on another plant), native to North America. While the remaining are monoecious (both male and female flowers are found on a single individual plant). The latter species are wide-spread across the different continents, from tropical lowlands to the Himalayas [7]. In the Aztec pre-colonial period, *Amaranthus* was a significant plant, as it was central in the worship of gods. It is also used as a medicinal plant, a source of dye, fodder, and an ornamental plant, in addition to its, uses as a staple food in places where they have been cultivated [8]. It is reported that *Amaranthus* has adapted to the sub-Saharan African region, such that it is a plant that grows even along the roadsides and are also seen growing on many fallow landscapes [9]. In South Africa, for example, *Amaranthus* is one of the most common indigenous leafy vegetables, which grows well in the summer season. To be precise, *Amaranthus* is found in places like KwaZulu-Natal, North West, Limpopo, and Mpumalanga, the homeland provinces of *Amaranthus*, where it is grown massively [9]. *Amaranthus thunbergii, A. greazicans, A. spinosus, A. deflexus, A. hypochondriacus, A. viridis,* and *A. hybridus* are the most predominant species. *Amaranthus* being a C4 plant, the afore-mentioned species are all tolerant to adverse climatic conditions even, in prolonged dry periods [10–12]. Although, *Amaranthus* is rarely cultivated when compared to other African leafy vegetables [10–12]. In countries where *Amaranthus* have been grown, various *Amaranthus* colors exist, ranging from gold, red, green to purple [1]. The majority of *Amaranthus* species have edible seeds and leaves, with some species known as vegetable amaranths, which include: *A. blitum, A. lividus, A. viridis, A. gracilis* Desf. and *A. tricolor,* which is synonymous with *A. tristis* L., *A. gangeticus* L. [13]. Several studies on the grain *Amaranthus* have been conducted while *Amaranthus* leafy vegetables are being relegated in research resulting in limited information on the potentials of the leafy vegetable of *Amaranthus* compared to the grain *Amaranthus* [14,15]. *Amaranthus* seeds are edible just as other cereals usually, it is commonly known as pseudo-cereal [4]. Several studies reported that *Amaranthus* seeds are a good source of gluten-free protein, which has been explored for making a variety of snacks across the world [16,17]. *Amaranthus* leaves have been used as greens, just like spinach. Also, it has been used as seasonings, just like mint being used in food. It has also been used in salads. Similarly, *Amaranthus* can be stewed with other vegetables like onions, garlic, tomatoes, they are sometimes used as dish like pepperpot [3]. Just as other vegetables are preserved, the leaves of *Amaranthus* 'shelf-life can be extended by drying and milling it into powder and, can be used for sauces preparation, which might improve its utilization even when not in the season [2,9,18]. Processing of *Amaranthus* leaves into leaf powder, as a preservative measure, especially in South Africa. has been overlooked and thus, further contributing, to it, being underutilized especially when it is out of season. On the other hand, it is reported that about ten species of *Amaranthus* are being considered troublesome weeds [19]. Despite people's perceptions, *Amaranthus* has received a resurgence in recent decades. Consistent studies show that *Amaranthus* is a multipurpose plant because the grains, stem, and leaves are edible, and the leaves could be utilized to enhance convenience food of low nutritional quality, especially

instant noodles [9]. Furthermore, *Amaranthus* can be explored as smart food for tackling malnutrition challenges especially the scourge hidden hunger. Even-though *Amaranthus* have been grown across most of the continents of the world, there is still scarcity of holistic information on the great potentials in *Amaranthus*, especially of leafy variety [4]. *Amaranthus* species are tropical climates plants. They can thrive well without pesticides and fertilizers since it belongs to the C4, dicotyledonous group of herbaceous plants [14]. The C4 plants have been known to use a specific type of photosynthesis mechanism known as (C4 photosynthesis) this is to avoid photorespiration, which helps the plants to grow better in hot, dry environments; photorespiration is a condition where plants close their stomata to conserve water [3]. This means as the stomata closes, the carbon dioxide levels in the interior of the leaf falls, and the oxygen levels rises [20]. The C4 plants have been identified to grow well under temperatures above 25 °C during the day, and at night the temperature must not be lower than 15 °C. The C4 plants have demonstrated the ability to survive with little attention. Bright light and adequate availability of nutrients can be beneficial for maximum yields [10].

*Amaranthus* has also been considered a superfood. Hence it is a promising and unique plant to be explored for its great intrinsic, essential nutrients since they are rich sources of micro-nutrients (essential vitamins and minerals), which have been studied as inevitable to optimum wellbeing. It is endowed with several compounds, including amino acids: lysine, arginine, histidine, leucine, cysteine, phenylalanine, isoleucine, valine, threonine, and methionine [21]. Two broad groups of *Amaranthus* exist the grain and leafy *Amaranthus*. Although the leaves of the grain *Amaranthus* group are also consumed however, in places like Asia, the *Amaranthus* plant is specifically grown for its grain purposes. In addition, *Amaranthus* that is specifically grown for leafy purposes is subdivided into two species: *A. tricolor* and *A. lividus*, which are both equally called *Amaranthus* and are endowed with medicinal and nutraceutical properties [22]. It is reported that the phytochemical extract from leaves of *Amaranthus* spp., such as *A. viridis* has biologically active components [9]. These active components include tannins, saponins, phenols, flavonoids, cardiac glycoside, steroid, and triterpenoids [21]. Likewise, *A. viridis* possesses some chemical constituents that exhibit potent activities like anti-inflammatory, antihepatotoxic, antiulcer antiallergic, antiviral actions [21]. For example, *A. viridis* has been used in places like India and Nepal in traditional medicine to reduce labor pain and to act as an antipyretic, which reduces fever [21]. Similarly, in Spain, specifically the Negritos of the Philippines, used *Amaranthus* leaves directly on skin diseases to cure eczema, psoriasis, and rashes. It has also been used as an anti-inflammatory agent of the urinary tract, venereal disease vermifuge, anti-rheumatic, antiulcer, analgesic, antiemetic, laxative, it improves appetite, is an antileprotic, for the treatment of eye problems and asthma [21]. Moreover, *Amaranthus* leaves are an important source of traditional medicine besides their use as a source of nutrition for humans and forage purposes. respiratory and

Although *Amaranthus* possesses many medicinal and nutraceutical potentials, it is a crop that has suffered neglect, which perhaps led to its being stigmatized as a poor man's food plant [4]. *Amaranthus* is perceived among the youth and urban population as a food plant for the poor [10]. However, on the contrary, among the wealthy, it is gaining interest and has been rediscovered as an important traditional leafy vegetable with several protective and curative properties, which are attributed majorly to the strong antioxidant and phytochemical properties [23,24]. Similarly, the systems of *Amaranthus* cultivation, processing, and value-adding techniques that could promote its utilization is now receiving research interest.

Currently, there is a trend in which people are health conscious of their diet, as health challenges are on the rise as well as taking in of food supplements, healthy foods, including calciferous vegetables, are being preferred [25]. However, these supplements and healthy vegetables are expensive and are not accessible to the less-privilege. Even though a variety of food plants, including *Amaranthus,* has been identified to supply most of the dietary needs, but more so, they are seen as cost-effective and sustainable food [8,26]. Lack of

sufficient information on its potential has limited its optimal utilization and maximum health benefits of *Amaranthus*. Such a dilemma is the drive for the interest in research on the potentials of traditional leafy vegetables like *Amaranthus*. Hence a comprehensive review of the nutritional composition and potentials of leafy vegetables, such as *Amaranthus*, is important, which is the focus of this study.

While the calciferous vegetables (cabbage, kale, cauliflower, broccoli, lettuce, spinach, asparagus, cilantro celery) are quite famous and may be preferred, *Amaranthus* popularity has received a resurgence, and it is currently considered among the superfoods with numerous health benefits.

## 2. An Overview Background of *Amaranthus*

Amaranthaceae is in the family name for all Amaranth, a genus *Amaranthus* spp. [4,24]. *Amaranthus* emanates from an ancient Greek word connoting (flower), meaning eternal or not wilting, unfading, or life everlasting [21,27]. *Amaranthus* belongs to a sub-family of Amaranthoideae [24,28]. Its domestication is dated back to the 6th-century BC, among Aesop's fables, who described *Amaranthus* as a short-lived flowering plant, having everlasting beauty when compared to roses. Phenotypically, *Amaranthus* has an attractively arranged inflorescence, a flowering plant that grows naturally or cultivated easily around home gardens. Research shows that about 60 to 75 species of *Amaranthus* with the family of Amaranthaceae are dispersed throughout the world, with only a few that are cultivated, and it originates from the temperate, subtropical, and tropical climate zones [13,27]. *Amaranthus'* real origin is traced to America, where it is recognized as an immortal, a staple crop in the Aztec, Mayan, during the Incan civilizations [24]. Currently, *Amaranthus* has spread widely across the globe. It is cultivated and consumed throughout India, Nepal, China, Indonesia, Malaysia, the Philippines, the whole of Central America, Mexico, and Africa [24]. Among the grain types, some species are considered native to South and Central America, while other types are native to Europe, Asia, Africa, and Australia [13,27].

In Western, Central, and South America, *Amaranthus* are mostly recognized for their grain purpose species. But there are cultivars grown for their leafy purposes. The leafy vegetable *Amaranthus* includes *A. tricolor*, *A. lividus*, *A. dubius*, *A. blitum*, and *A. hybridus*, which are crops of Africa, Southeast Asian, and Central American origin [29]. *Amaranthus* was cultivated by the mighty Aztecs 6000–8000 years ago in central Mexico, where it was not only a staple food for the Aztecs, though it played a vital role in the Aztecs' worship of gods and the Aztec human sacrifice rituals [4,8]. During the Aztecs period, statues of their gods were built from the mixture of *Amaranthus* grain and honey. After the worship of the gods, the status is broken and is distributed to people for eating. Hence, the practice retard *Amaranthus* as a staple food [30]. This happened on the arrival of the Spanish with Cortez, as Christianity was being forced on the Pegan natives. Thus, the grain was banned, and the fields were burned, forbidden, cultivators or possessors of *Amaranthus* were severely punished. However, they were unable to destroy and eradicate the grain [30]. Despite *Amaranthus* 'rough history, little amounts of its grains managed to survive in a few remote areas, where the survived grains were primarily used for making traditional sweet called *Alegria* [30]. *Amaranthus* is currently grown in other parts of the world with few locations in the United States, Asia, and across Africa. Evidence of the cultivation of *Amaranthus* seeds emanates from the Coxcatlan cave in the Tehuacán Valley of Mexico, traced early to 4000 BC. Thereafter, evidence, such as burned and black *Amaranthus* seeds, was found throughout the US Southwest and the Hopewell culture of the US Midwest. Then, the *Amaranthus* grain was re-introduced in the United States in the 1970s [30]. Long after 300 years of it, being a less important crop, *Amaranthus* was rediscovered in Mexico, where it was shared on a ceremonial day with the descendants of the Aztecs, who believed that *Amaranthus* provided them with supernatural power in their religious practices. Even though *Amaranthus* is regarded as a neglected/lost crop [31], *Amaranthus* is well-known in many rural and urban communities because it is utilized at the household level; however, the mode of consumption differs by ethnic group. Although, *Amaranthus* appears to be a

lost crop to many researchers and policymakers across the world, however, in localities where they have been cultivated and are appreciated, *Amaranthus* have provided humans with food. Nevertheless, updated information on their potentials and uses is needed as this can be informative and contribute to providing solutions to the world's threatening problems of food insecurity and malnutrition challenges.

### 3. *Amaranthus* a Food Solution for Food and Nutrition Insecurity

There has been global progress in reducing nutrient-related challenges, especially in developing countries [32]. However, the difficulties that seem to hinder this achievement still include hunger, climate change malnutrition, including rural poverty, and lack of nutrition education, which affects the nutrition security of many populations, as many are unable to make wise choices of the available nutrient-dense food ingredient option that can enhance the inadequacies in most staple crops [33]. Vegetables, including *Amaranthus*, have been identified as active ingredients that can contribute to the food supply of human essential dietary needs that can tackle hidden hunger issues. Although varieties of exotic vegetables like kale and collard greens, linked with high status, have been investigated as food solutions to nutrition security, they are inaccessible to the common man. Across the literature, *Amaranthus* is often linked with poverty. Hence, people do not want to be identified with poverty even though they may be poor [4]. Such individuals lack the education that traditional vegetables, including *Amaranthus* variety, are being promoted as a promising plant with essential minerals and vitamins, nutraceutical, and phytochemicals properties [3,31]. Though the exotic vegetables appear to provide a sustainable measure for improving global food availability and food and nutrition security, little attention is shown to vegetables that are considered "lost crops", including *Amaranthus* [34].

Generally, several agricultural researchers and nutritional experts argue that vegetables, including *Amaranthus*, are endowed with essential nutrients yet to be exploited. For example, *Amaranthus* is currently ascribed worth as a superfood. Therefore, exploring traditional varieties of vegetables like *Amaranthus* for food and nutrition security cannot be overemphasized because, the nutritional needs of humans could be met via its inclusion in staple foods, which is a cheaper solution to preventable food-related health challenges incapacitating the human potentials.

It is believed that *Amaranthus* is not a true cereal, like wheat, sorghum, millet, maize, or barley, but, somewhat, it is considered as "pseudo-cereal" like the buckwheat crop [4,15].

Pseudo-cereal is one of the non-grass plant's crops that have been used, like any other true cereals belonging to the grass family. Hence, *Amaranthus* seeds can be grounded into flour and can be used in baking varieties of food products, especially snacks. Although *Amaranthus* are commonly known as pigweed, it is a unique plant that grows on its own. Hence, they are broadly cultivated world-wide for many reasons of interest, including industrial, medicinal, ornamental, fodder, and nutrition purposes. *Amaranthus* is considered a multipurpose plant since it has proved its worth in supplying cereal grains and leafy vegetables of high essential nutritional value for animals and human nutrition.

The essential nutrients found in *Amaranthus* include protein, calcium, iron, vitamins A, C, and K, riboflavin (B2), niacin (B3), vitamin B6, and folate (B9). Several studies attest that *Amaranthus* benefits are enormous and undeniable. Therefore, a regular diet plan can help combat nutrient-related health issues; malnutrition and especially micronutrient deficiency challenges [9,27,35].

However, *Amaranthus* is a common crop that has been given little attention in research, though it is identified as a promising plant capable of combating most of the human nutrient deficiency challenges. Interestingly, *Amaranthus* supplies nutraceutical and phytochemical properties and other dietary needs of humans, though *Amaranthus* has been identified as low in calories, which can be a good option for weight loss. *Amaranthus,* being one of the superfoods plants, the leaves have been accessible in fresh form, but the powdered form of it is still scarce. This implies that the processing aspect of *Amaranthus* leaves into a new form or product, such as powder, is lacking or still at a low level. Although

researchers still recommend the powder as having great potentials and are cost-effective, sustainable for alleviating hunger, but it is also vital for food and nutrition insecurity problem, especially among the vulnerable group because it could be used when it is out of season [36]. Although *Amaranthus* is drought-tolerant, nutrient-dense, studies attest that it is fizzling out in some regions of the world where it was domesticated hence, making it an under-explored plant. Moreover, while malnutrition is still staggering, it continues to impact the burden of diseases and mortality on the vulnerable within the population of many developing countries, including sub-Saharan African countries. Hence, more interest in research around Amaranthus is needed to explore its potential and uses as viable solutions to food and nutrition insecurity. In addition, a novel medium of its delivery of the nutritional potential could be done via its inclusion of any of the *Amaranthus* species into most of the staple foods.

### 4. Common Species of *Amaranthus* Grown for Grain and Leaves and Their Uses

Botanically, *Amaranth's* species are not true cereals, like wheat and rice. However, they belong to the pseudo-cereals class. They are grown in many regions of the world for their edible leaves and seeds as the two main categories [2]. Primarily, the three species of *Amaranthus* grown across the globe for grains include *A. cruentus*, *A. caudatus*, and *A. hypochondriacus*. Conversely, this does not mean their leaves and stems are not consumed. They are consumed when they are still at a tender stage. *A.cruentus* has been reckoned among crops with a high protein content that contains essential amino acids, including methionine and cysteine [2,37]. The flour of the seed has been added to other cereals for making healthy snacks and complementary foods [2,9]. *Amaranthus* leaves can easily be prepared and consumed as a vegetable dish or as an ingredient in a sauce [2]. The tender stems and leaves can be steamed or stir-fried and served as a side dish. It can also be cooked with assorted meats and fish in various oil, such as groundnut and palm oil [2]. of *Amaranthus* differs from the localities where they are grown. It is usually not consumed alone, but it is also served as an accompanying dish. From the literature, Consumption there is no clear separation between vegetable and grain species. The reason is, the tender leaves of the grain varieties are equally utilized as the leafy *Amaranthus* [2]. In Eastern and West Africa, for example, in Nigeria precisely, *Amaranthus* is a common vegetable that complements most carbohydrate dishes, such as pounded yam, *amala, tuwo,* and *fufu.* It is also recommended to boost one's red blood count [38,39]. The popular leafy *Amaranthus*, include *A. cruentus*, *A. dubius, A. blitum*, and *A. tricolor*. *A. cruentus* has been cultivated both as a vegetable, fodder, medicine, and grain depending on the targeted market of the producer [2].

Different Amaranthus species (used as leafy vegetables, grains, ornamental and medicinal purposes) that are unique to some regions of the world have been identified. For instance, Figures 1 and 2 below describe *Amaranthus*, species, status, and their uses. Figure 1A describes *Amaranthus dubius* or wild spinach with palatable flavors hence, they are consumed as leafy vegetables foods for human and animals and are used for medicinal purposes (1B) *Amaranthus viridis*, are cultivated as leafy vegetables and for medicinal purposes. Even though *Amaranthus viridis*, are wild, they are occasionally cultivated as leafy vegetables, fodder, and for medicinal purposes [4]. Figure 2A–C describes common species of wild leafy *Amaranthus* varieties of *A. spinosus,* used as leafy vegetables for fodder and medicinal purposes. *Amaranthus* classification is described in Table 1, while the common names for *Amaranthus* in different countries are shown in Table 2. The vegetable *Amaranthus* are easily known by their inflorescence features, mostly axillary glomerulus, or short spikes. The flower originates from the bud of the leaf axil, three tepal lobes, three stamens, brownish-black seed with an indeterminate kind of growth [4]. The grain *Amaranthus* are categorized by apical large to moderately large complex inflorescence containing aggregates of five stamens, cymes, five tepal lobes, seed with variable seed color, and well-defined flange utricle circumscissile [4]. Furthermore, grain *Amaranthus* is characterized by having discoid grains with a well-differentiated folded flange region and

seed coat color other than black or brownish. However, some of the weedy *Amaranthus* that are cultivated can be used as vegetables. The weeds species are similar in their morphological attributes with the leafy vegetable form, compared to the grain *Amaranthus*. Seed character is vital in differentiating the vegetable, grain, and weed *Amaranthus* [4].

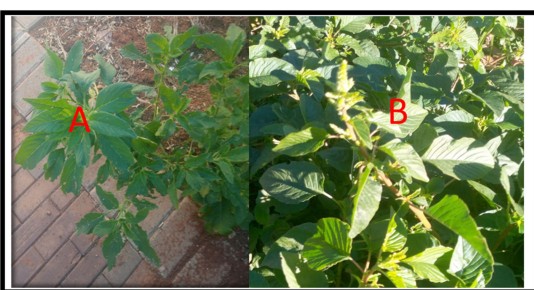

**Figure 1.** (**A**) *Amaranthus dubius* or wild spinach (**B**) *Amaranthus viridis*.

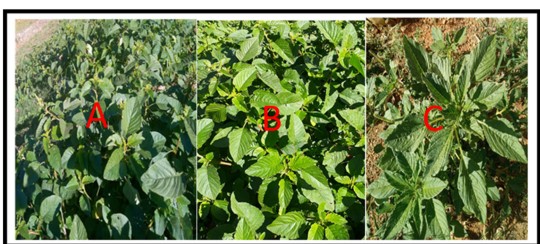

**Figure 2.** (**A**–**C**). Variant of *Amaranthus spinosus*.

**Table 1.** Classification and description of *Amaranthus* [4,40].

| Vegetable *Amaranthus* | Grain *Amaranthus* | Weed *Amaranthus* |
| --- | --- | --- |
| *Amaranthus tricolor* | *Amaranthus cruentus* | *Amaranthus spinosus* |
| | | *Amaranthus hybridus* |
| var. *tricolor* | *Amaranthus hypochondriacus,* | *Amaranthus viridis,* |
| *Amaranthus tricolor* var. *tristis* | *Amaranthus caudatus* | *Amaranthus retroflexus* |
| | | *Amaranthus graecizans* |
| | | *Amaranthus dubius* |

There are no clear differences between vegetable and grain species of *Amaranthus*. This is because the leaves of young grain varieties are also used as pot herbs sauce. Aside from the wild (*A. spinosus*) described in Figure 2, grows voluntarily in fallow landscapes, four popularly cultivated species of vegetable *Amaranthus* exist: which include: *A. cruentus, A. dubius, A. blitum,* and *A. tricolor.* However, it is was identified that *A. cruentus* can be cultivated both as a vegetable and as grain, depending on the target market of the producer [2]. This occurs because the weeds species show morphological commonality with leafy vegetable form, other grain forms of *Amaranthus*. The seed character is very useful in differentiating vegetables, grain, and weed *Amaranthus*. This because the vegetable group shares a striking similarity with the weed group in having brownish-black or black seeds.

**Table 2.** Common names of Amaranth in different countries of the world.

| Number | Language | Vernacular Name | Author |
|---|---|---|---|
| 1–5 | South Africa: Afrikaans | Hanekam, kalkoenslurp, misbredie, varkbossie | [41] |
| | Tswana | Imbuya, thepe | [41] |
| | Venda | Umfino, vowa, Morogo | [41] |
| | Xhosa | Umfino, umtyuthu, unomdlomboyi | [41] |
| | Zulu | Imbuya, isheke | [41] |
| 6 | Congo | Bitekuteku *Amaranthus viridis*, Kinshasa Province | [41] |
| 7 | Indonesia | Bayam | [27] |
| 8 | Laos | Pak hom | [27] |
| 9 | Siri Lanka | Thampala | [27] |
| 10 | India | Rangasak, ramdana, rajeera, lalsak, lalsagchauli; cheera; koyagura; kuppaikeerai; thotakura | [41] |
| 11 | China | Een choy, Yin choy, In-tasi, Hsien tasi, xiancai, Hiyu, Hon-toi-moi, | [41] |
| 12 | Japan | Hiyuna | [41] |
| 13 | Spanish | French: calalou, callaloo | [41] |
| 14 | Fulani | Boroboro | [41] |
| 15 | Ghana | Madze, efan, muotsu, swie | [41] |
| 16 | Sierra Leone: Grins | Creole, hondi, Mende | [41] |
| 17–19 | Nigeria: Hausa | Alayyafo | [41] |
| | Yoruba | Efo tete, Eforiro | [41] |
| | Igbo | Igbo inene; Temne: ka-bonthin | [41] |
| 20 | Malawi: | Bonongwe | [41] |
| 21 | Philippines | KulitisIlongo, uray Tagalog | [41] |
| 22 | Indonesia: | Bayam, Bayammenir, Java, Bayamkotok Sumatra, Chaulai | [41] |
| 23 | Thailand: | Pak-komhat, pak. Phomsuan | [27] |
| 24 | Jamaica: | Callaloo | [41] |
| 25 | Vietnam | Yan yang | [27] |
| 26 | Peru | Anchita, achos, achis, incajtaco, coimi and kiwicha | [27] |
| 27 | Bolivia | Coimi, Millmi | [27] |
| 28 | Ecuador | Sangoracha, alaco | [27] |
| 29 | China | Hiyu, hon-toi-moi, yin choy, hin choy, een choy, tsai | [27] |
| 30 | India | Chhaulai, Rangasak, ramdana, rajeera, lalsak, lal sag | [41] |
| 31 | Malaysia | Bayamputeh, bayarnmerah | [41] |
| 32 | Caribbean | Spinach, bahaji callaloo, calaloo, etc | [3] |
| 33 | English: | Prickly amaranth, needle burr, spiny amaranth, thorny amaranth, pigweed, African spinach, foxtail | [41] |
| 34 | Hindi: | Kantachaulai, Gujarati: Kantalodhimdo, Kantanudant. Manipuri: Chengkruk Marathi: | [41] |
| 35 | Tamil: | Mullukkeerai, Malayalam: Kattumullenkeera Mullatotakura Kannada: Mulluharivesoppu | [41] |
| 36 | Telugu: | Bengali: Kantanotya Oriya: Kantaneutia Sanskrit: Tanduliuyah | [41] |
| 37 | French | amarante; brede de Malabar, queue de renard, discipline des religieux | [41] |
| 38 | Mexico | Zac-tec | [41] |
| 39 | Portugal | Caruro | [41] |
| 40 | Sweden | Mchicha | [41] |
| 41 | America | "Chowlai" | [21] |
| 42 | Philippines | Kulitis | [27] |
| 43 | Thailand | Pakkhom ha, pak, khomsuan | [27] |

## 5. Potential Nutritional Value and Health Benefits of *Amaranthus* Leaves

The production of Staple food crop appears unable to meet up with the required food that feeds the increasing global population. Similarly, staple foods are inadequate in essential nutrients to optimize wellbeing [42]. Hence, there are increased studies on food that are easily grown, and are nutrient-dense in composition to tackle hunger, mitigate and alleviate food and nutrition insecurity [27]. *Amaranthus*-based food can be a contributor to food and nutrition security, especially among the less privileged, who are unable to access animal protein and other essential nutrient food sources like calciferous vegetables. This is because *Amaranthus* leaves are reported as one of the plants rich in essential nutrients among other traditional vegetables, of great importance, besides its use for religious practices in the place of its origin [4,43]. *Amaranthus* has been known to have relatively high nutritional value compared to exotic vegetables that are mostly preferred. However, it has been established that the contribution of the micronutrient status of *Amaranthus* depends upon their retention capacity of the nutrients after processing and cooking methods [44]. *Amaranthus* leaves are rich in protein and micronutrients such as zinc, calcium, magnesium, phosphorus, folic acid, potassium, iron, and vitamins A–C but low in carbohydrates [3,45]. *Amaranthus* leaves are low in calories. Consequently, they must be complemented with foods rich in calories. *Amaranthus* is cholesterol-free, making it a food option for those on strict diet prescription, such as weight losing formulation. A study confirms that the nutrient content per 100g fresh weight of *Amaranthus* was compared to cabbage and was reported higher in protein, calcium, iron, β-carotene, and vitamin C [3]. Likewise, lettuce was compared with *Amaranthus*, and *Amaranthus* showed a higher nutritional content of 7 times more iron, 13 times more vitamin C, 18 times more vitamin A, and 20 times more calcium than other similar plants [43]. Since *Amaranthus* yields significantly within the shortest period, it could be a promising crop to tackle food security and food and nutrition insecurity [46]. Additionally, *Amaranthus* leaves are rich in antioxidants and phytonutrient properties. Thus, they are endowed with an essential dietary nutrient that could be explored as a preventive measure and healing effects to the essential nutrient-deficient individual [47]. The common names of Amaranthus in South Africa and other countries are described in Table 2.

*Amaranthus* has been demonstrated to have medicinal effects in traditional medicine [13]. For instance, the dried *Amaranthus* crop has been burned in Benin for the preparation of potash. Similarly, the root of *Amaranthus* has been boiled with honey in Senegal and was used as a laxative for infants. The water of macerated *Amaranthus* plants has been used in Ghana as a wash to treat pains in the limbs. Ethiopians have used *A.cruentus* as a tapeworm expellant [48]. The ash from *Amaranthus* stems has been used for wound dressing in Sudan, also in Gabon. Moreover, *Amaranthus* has been suggested to assist people with low red blood cell count [27,48]. The leaves of *Amaranthus* have medicinal properties that are beneficial to the health of young children, lactating mothers, and patients with constipation, fever, hemorrhage, anemia, or kidney complaints. Amaranth is rather included among the diuretics (substances that promote increased urine production) [48]. *Amaranthus* leaves have been used as anti-inflammatory [48]. The ability of *Amaranthus* vegetables to promote health benefits is largely attributed to the nutritional and bioactive compounds endowed in *Amaranthus*. For instance, studies on *Amaranthus spinosus* have been investigated, and a wide and spectrum of its pharmacological actions have been explored [41]. The plant possesses hepatoprotective, antioxidant activity, water extract of the plant have a significant immune-stimulating activity, and extracts had been used as an antidiuretic, antiviral, antimalarial, antibacterial, anti-inflammatory, antimicrobial, and hepatic disorders [41]. It is used internally in the treatment of internal bleeding, diarrhea, and excessive menstruation [41]. *Amaranthus* has not only been of benefits in traditional medicine, ornamental and industrial benefits but also, *Amaranthus* has widely been investigated for its nutritional composition, and have been reported to have potentials in human and animal nutrition; hence, they are identified as rich in the following dietary nutrient [4,49].

Fiber: *Amaranthus* has been considered a rich source of fiber, which constitutes soluble and insoluble [50,51]. Therefore, the inclusion of *Amaranthus* leaves as one of the ingredients in a daily diet can be of advantage to the consumers, as it aids digestion and as well as beneficial, by way of easing bowel movement, managing weight and thereby, reducing non-communicable diseases, especially heart disease and diabetes. High protein and fiber components of *Amaranthus* can assist in reducing appetite, thereby promoting individual weight loss and healthy living [50,52]. Due to the fiber content, *Amaranthus* digest easily; hence, it can be beneficial to convalescents and those recovering from illnesses. In addition, its leaves have been known to treat diarrhea and hemorrhages.

Protein. Dark green leafy vegetables, including *Amaranthus* leaves and the grain, are rich sources of plant protein [53]. *Amaranthus* is among the vegetables that are rich in essential amino acids [54]. Protein from the plant has been regarded as much healthier protein than protein from the animal. This is because plant sources are having less or no fat and, therefore, cholesterol-free. It has been noted that bad cholesterol in the body leads to cardiac problems. Amaranth leaves are known to lower bad cholesterol, which is responsible for many cardiac problems.

Also, it is claimed that consumption of *Amaranthus* leaves suppresses appetite since they are rich in protein [3]. A protein-rich diet can suppress hunger because it reduces insulin levels in the blood, thus controlling the appetite. *Amaranthus* is gluten protein-free. Thus, it can be beneficial to those who are gluten intolerant. Proteins are macronutrients required by humans for the growth and maintenance of the human system. It plays a critical role in regulating different activities at the cellular, tissue, and organ levels of the body, providing the right hydrolytic environment [55]. Plant and animals are rich sources of protein. Plant proteins form a large part of the human diet. However, deficient in one or more essential amino acids are regarded as incomplete protein. If certain food supplies enough of seven out of the eight essential amino acids, the lacking amino acid is referred to as the "limiting amino acid". These limiting proteins (lysine, threonine, methionine, and tryptophan) could be found in plants, such as *Amaranthus*. Hence, *Amaranthus* has been identified to have lysine in a significant amount. Moreover, lysine has been known to help to promotes hair growth and slows aging processes and enhances good skin. In addition, people who suffer hair loss or the early graying of hair can benefit maximally by including *Amaranthus* leaves in their diet. In addition, lysine is an essential amino acid, which is essential for energy production and a good precursor for the absorption of calcium. Both *Amaranthus* leaves and the grain is a rich source of protein. Currently, *huaútli* is a superfood, and it is gaining worldwide recognition as a high-protein edible plant with great potentials to provide a solution for the world's hunger and to tackle malnutrition challenges [4]. The seeds of *Amaranthus* have eight to nine grams of protein in a one-cup serving, offering a nutritionally complete plant food with all the essential amino acids needs of the human body, without gluten.

Iron: *Amaranthus* is rich in iron [24,56,57]. Iron is required for producing red blood cells and is also needed for cellular metabolism. However, iron found in *Amaranthus* leaves can be maximized through cooking methods like the blanching method of food processing, which has been known to improve the bio-availability of iron [58]. In addition, iron could be made available with the presence of vitamin C-rich foods or supplements in the diet. This is because vitamin C food sources such as lemon, apples, or orange, have been proven to facilitate maximum absorption of iron in the blood [58,59]. This implies that maximum iron absorption and can help fight anemia. In addition, iron-rich foods, such as *Amaranthus*, can be explored to combat iron deficiency challenges. Hence, it is a source of immunity boosters to the world where the iron problem prevails.

Vitamin C. Leafy green *Amaranthus* are rich in vitamin C. It is reported that one cup of boiled and drained *Amaranthus* leaves contains 90% vitamin C daily dietary requirement [4]. Vitamin C is a water-soluble vitamin, and it is essential and potent to fight infections as well as enhance the quick healing of scurvy and other wounds. Besides vitamin C, phenolics and flavonoids are phytochemical compounds responsible for most

of the antioxidant activity in fruit and vegetables; hence, they are identified as essential for the biosynthesis of collagen, carnitine, which is a metabolism booster and a source of antioxidants and anticarcinogenic compound [49,60]. These compounds occur as secondary metabolites that act as defenses against several diseases like atherosclerosis, including cancer, arthritis, cataracts, emphysema, retinopathy, neurodegenerative and cardiovascular disease [61]. As a result, there is increasing interest in the exploitation of plant-based drugs. The plant-based drug is considered a safe and cost-effective alternative to other therapy [47]. Therefore, the inclusion of *Amaranthus* leaves as a regular part of a diet is a promising nutraceutical therapy that can be explored as a preventive mechanism against non-communicable diseases ravaging people's life.

Vitamins are organic compounds, having essential nutrients that cannot be synthesized by the body; hence, they must be derived from the diet consumed, including *Amaranthus* leaves diet [62]. Essential vitamins and minerals are good boosters of immunity [63]. Several studies have linked diets rich in vegetables like *Amaranthus* to have the potential to reduced risks of chronic diseases or vitamin deficiency syndromes [62,64]. Similarly, studies report that a healthy lifestyle and ample consumption of vegetables and fruits are linked to lowering non-communicable diseases, including heart diseases [8]. Hence, it is generally agreed that people who consume adequate fruits and vegetables are usually healthier and free from hidden hunger and its consequential issues. Fruits and vegetables, including *Amaranthus* are good sources of vitamins and minerals.

Vitamin A. It is reported that *Amaranthus* leaves are endowed with vitamin A properties amounting to 73% of dietary needs. A cup of *Amaranthus* leaves has been proven to supply 97% of daily human dietary needs [4].Vitamin A is essential for several processes in the body. It is good for the maintenance of a healthy vision. It aids the functioning of the immune system and organs, as well as aids the proper growth and development of babies in the womb [65].

Vitamin K. Vitamin K is a fat-soluble vitamin. It occurs naturally in two forms; phylloquinone (vitamin K1) and menaquinone (vitamin K2) [66]. It has been reported that phylloquinone levels in fruits and vegetables range from extremely low to quite high [62]. Green leafy vegetables, including *Amaranthus* leaves, have been reported to have the highest amount of vitamin K [62,67]. Vitamin K is required for wellbeing because it plays an important role in blood clotting and also essential for healthy bones because it promotes osteoblastic activity and strengthens the bone mass [62]. Also, vitamin K has been known to control neural damage in the brain. Hence, it is beneficial for those who suffer from Alzheimer's disease. Vitamin K as a cofactor in the carboxylation of osteocalcin has been identified to help to improve its affinity for calcium and promote binding to hydroxyapatite, thereby improving healthy bones. [68].

Calcium. *Amaranthus* leaves have been reported as a good source of calcium. A cup of *Amaranthus* leaves provides a significant (28%) amount of calcium [4,69]. Calcium is an essential mineral that can be beneficial to humans especially, people who have osteoporosis and other calcium deficiency-related health problems.

Potassium. *Amaranthus* has been investigated for its mineral content and was reported to contain significant content of potassium [43,70]. Potassium is an essential nutrient largely found in body cells. It is the main electrolytes that play an important role in the living cell. It maintains fluid balance. Hence, it is crucial for muscle contraction, nerve conduction as well as providing acid-alkaline balance in the body [71]. Adequate intake of potassium can contribute to the maintenance of healthy blood pressure and heartbeat. However, it has been reported that the low intake of dietary potassium plays a role in the development of cardiovascular diseases, including high blood pressure [72]. Hence, the adequate intake of dietary potassium from vegetables like *Amaranthus* may contribute to prevent and reverse high blood pressure, provided there are no other underlying health challenges. However, such a dietary lifestyle needs to be consumed along with more fruits and vegetables as good sources of essential vitamins and minerals because fruits and vegetables have been reported as good sources of potassium as opposed to sodium [67,73]. Traditional leafy

green vegetables like *Amaranthus* have been identified as a super-food that can provide the body with essential vitamins and minerals vital for optimum wellbeing [8]. The intake of dietary nutrients including potassium, has been reported low among the under-privileged population [74]. White fruits and vegetables, such as (bananas, white peaches, parsnips, cauliflower, garlic, mushrooms, onions, potatoes, shallots, turnips, corn, and kohlrabi) have been reported as rich sources of fiber, magnesium, and potassium. However, they may be expensive for the under-privileged population [75]. However, vegetables like *Amaranthus* are cheap yet with great potentials for essential nutrients, including potassium. Moreover, *Amaranthus* could be naturally grown since it is easily cultivated with little attention.

The B vitamins. *Amaranthus* leaves are rich sources of vitamins B complex group: riboflavin, niacin, folates, thiamin, vitamin B6, vitamin B1 (thiamin), vitamin B2 (Riboflavin), vitamin B3 (niacin), vitamin B6 (pyridoxine), vitamin B9 (folates), vitamin B12 (cobalamins cyanocobalamin) and others [8]. The B vitamins are beneficial as they are needed for optimal mental and physical health as well as help prevent birth defects in newborn babies [8]. B vitamins are regarded as cofactors for enzymes that are involved in the energy-producing metabolic pathways for carbohydrates, fats, and proteins. B vitamins also play an important role in maintaining functions of the nervous system [76]. Vitamin B9, better known as folate, plays a central role in the one-carbon metabolism in the nucleotide synthesis, in the metabolism of homocysteine. However, the bioavailability of this plant also crucial to their maximum benefits to humans

### 6. Bioavailability of Micro-Nutrients in *Amaranthus* Leaves

Several studies reported that plant foods, including *Amaranthus* and other dark leafy vegetables, contain some levels/concentration of antinutritional compounds like phytates, oxalates, and nitrates, otherwise call antinutritional factors [77]. These compounds are believed to be present in a variety of plant food, and it has been reported that they are capable of reducing the absorption of micro-nutrients hence, leading to the lowering or rather hindering the bioavailability of essential nutrients in plant food [55]. Some of the antinutritional factors that interfere with the absorption of some vital nutrients in the diet include phytic acid, oxalates, pro-anthocyanidin, tannin, and dietary fibers, which are known to reduce the bioavailability of the nutrients after consumption [78]. Among other vegetables, *Amaranthus* has been considered a rich source of micro-nutrients and other dietary minerals. *Amaranthus* is considered as a store house for potassium, calcium, particularly iron, zinc, magnesium with an appreciable amount of carotenes and vitamins A–C, which have been investigated for optimum wellbeing [55]. For example, a study report that raw amaranth is considered rich in micro-nutrients, particularly iron and vitamin C. [78]. *Amaranthus* was compared with common vegetables, such as cabbage and spinach and was rated higher in terms of the nutritional composition [3]. Nevertheless, the bioavailability of the micro-nutrients in *Amaranthus* depends on their methods of food preparation, consumption, as well as the biological interactions between their phytonutrients content in *Amaranthus* and the nutrients available in added ingredients. The stability of beta-carotene, for example, was said to be more enhanced with vitamin C, lutein, polyphenols, and lycopene when interacted [60]. This implies that when *Amaranthus* is combined with food rich in these compounds, the concentration of beta-carotene will be greater since *Amaranthus* is being rediscovered, as plant crop of great importance. The most widely grown species Amaranthus, their synonyms and common names are described in Table 3 below.

**Table 3.** Some of the most widely grown species and their common name.

| Species | Synonyms | Common Name | References |
|---|---|---|---|
| *Amaranthus blitum* L | *A. lividus* L. <br> *A. oleraceus* L. | Amaranth, wild slender amaranth, pigweed, purple amaranth, amarantesauvage, amaranteblette, amaranto, bredo, (Po) mchicha, (Sw) | [79] |
| *Amaranthus curuentus* | *A. paniculatus* L. *A. sanguineus* L. *A. hybridus* L. subsp. *cruentus* (L.) Thell. var. *paniculatus* (L.) Thell. | Amaranth, African spinach, Indian spinach, amarante, brède de malabar, amaranto, bredo, mchicha <br> Red/purple amaranth | [80] |
| *Amaranthus caudatus* | Love-lies-bleeding, red-hot cattail, | African spinach, Indian spinach, brèdemalabar, Bredo, mchicha, | [81] |
| *Amaranthus dubius* | *Amaranthus tristis* auct. Non-L. | Spleen amaranth, pigweed, amarante, brède de malabar, <br> Amaranto, bredo, mchicha, | [82] |
| *Amaranthus hypochondriacus* | *Ahybridus* auct. Non-L. | Prince's feather, amaranth, amarante, brèdemalabar, amaranto, bredo, mchicha, prince-of-wales feather | [83] |
| *Amaranthus spinosus* L. | | Spiny amaranth, prickly amaranth, spiny pigweed, amaranteépineuse, épinardmalabar, épinard piquant, amaranto, bredo, mchicha, | [84] |
| *Amaranthus thunbergia* | | Wild amaranth, wild spinach, pigweed, amarantesauvage, amaranto, bredo, mchicha, Thunberg's *Amaranthus* | [85] |
| *Amaranthus tricolor* | *A. tristis* L. <br> *A. gangeticus* L | Amaranth, Joseph's coat, amarante, brède de malabar, <br> Amaranto, bredo, <br> Mchicha, Chinese spinach, fountain plant, tampala, summer poinsettia | [86] |
| *Amaranthus viridis* | *A. gracilis* Desf. Ex Poir. | Green amaranth, local tete, african spinach, Amaranteverte, épinard vert, épinard du Congo, Amaranto, mchichacalalu, slender amaranth | [87] |

## 7. Indigenous Knowledge System and, Uses of *Amaranthus* in America, Asia, and Africa

Indigenous knowledge can gradually disappear due to a lack of adequate transfer of knowledge to the younger generation. The loss of indigenous knowledge system of food may hinder people from benefitting from the available and accessible food resources thus, depriving people of the nutritional value embed in indigenous food, including *Amaranthus* [88]. Hence, conserving the indigenous knowledge system becomes a vital drive for food-based approach. This review agrees with the statement that, transfer of indigenous knowledge system of traditional vegetables is promising and can promote the accessibility of their endowed nutrient. *Amaranthus* is a common food plant that has been used in resource-poor rural populations. Hence it is linked with the poor [89].

Physiologically, *A. spinosus* is a straight plant, a perennial herb with diverse colors, and is widely distributed throughout the tropics and warm temperate regions of Asia from Japan to Indonesia, India, the Pacific islands, native to tropical America and Australia as a weed and as a cultivated crop. It is a common weed found in waste places, roadsides, and path sides and near rivers in West Africa, Ghana [89].The plant has a long history of usage in traditional medicine against various ailments around the world, subtropical and Himalayan regions, and is distributed in lower to middle hills (3000–5000 ft) of the entire Northeastern Himalayas [89]. *A. spinosus Linn.* One of the medicinal plants of Eastern Himalaya, especially of Sikkim Himalaya, is known as prickly *Amaranthus* [89]. Amaranths leaves and grains are indigenous foods of Incas and Aztecs in pre-Columbian times. Among the Aztecs, Amaranth flour was used to baked images of their patron deity, Huitzilopochtli, more prominently during the festival called *Panquetzaliztli,* symbolizing means "raising banners" The Mixtec's of Oaxaca also greatly recognized the role of Amaranths. Amaranths were made into a paste and were used to stick the precious postclassic turquoise mosaic covering of the skull, it was also used as a form of tribute payment. Its name in Nahuatl was *huauhtli* [89].

Furthermore, all *Amaranthus* has been used for forage since it was regarded as weeds, but it also has been included in the human diet as an ingredient with great potential

for wellbeing. *A. caudatus* was a commonly dispersed staple food of hunter-gatherers in South America and India. The species originated as one of the staple foods of the ancient inhabitants of the Andean region [89].

Several *Amaranthus* crops are widely grown in Central and South America; however, there are also grain amaranths that are genetically distinct from the weedy species, which are believed to have evolved from wild populations [90].

*Spinosus* Linn of the family of Amaranthaceae is traced to tropical America. A. spinous is spiky in appearance, commonly called "pigweed" [4]. *Amaranthus spinosus* is also one of the most common species of plant in India. Amarnthusspnous is generally known for their therapeutic properties "Kate waliChaulai (Kanatabhajii)"in Hindi [41]. *Amaranthus spinosus* is cultivated in India, Sri Lanka, and it spread across the tropics of most continents of the world (Central America, South America, Caribbean, Africa, and the warm temperate regions of Asia) from Japan to Indonesia, the Pacific Islands, and Australia where it grows as a volunteer crop or weed although it was also cultivated. Besides the supply of food, *Amaranthus* spinous can tackle the health challenges of malnutrition while optimizing wellbeing. The healing powers of traditional herbal medicines, including *Amaranthus* have been realized since ancient times. It is reported that almost 65% of the world populations have access to traditional medicinal plant and their knowledge system.

*Spinosus* amaranths have erected spikes, a yearly or perennial herb with diverse colors ranging from green to purple. In India, spiny *Amaranthus* are used for food and play a vital role in the Indian traditional system of medicine (Ayurveda). The plant has been used as febrifuge, antipyretic, laxative, and diuretic. *Amaranthus* spinous is used for treating digestible, bronchitis, used to stimulate appetite, increased bile secretion, it promotes lactation, increases hemoglobin, stomach flatulence, anorexia (restricting oneself from eating for fear of gaining weight), blood diseases, burning sensation, *leucorrhoea* (yellowish discharge from the female genital organ, leprosy, and piles). It is reported that *Amaranthus* spinous is endowed with phytochemicals, which are capable of destroying free radicals while boosting immunity, thus stressing its importance as a medicinal plant with great potentials [41,91]. *Amaranthus* contains active biochemical compounds that are reported in several studies. It is a rich source of alkaloids, flavonoids, glycosides, phenolic acids, steroids, amino acids, terpenoids, lipids, saponin, betalain, b-sitosterol, stigmasterol, linoleic acid, rutin, catechuic tannins, and carotenoids. Further studies on *Amaranthus* spinous have been conducted by various researchers, and a wide range of its pharmacological actions have been explored, which may include antidiabetic, antitumor, analgesic, antimicrobial, anti-inflammatory, and anti-spasmolytic a bronchodilator, hepatoprotective, spermatogenic, antifertility, antimalarial, antioxidant properties [41]. *Amaranthus spinosus* also use in internal bleeding, diarrhea. The next section presents several studies on *Amaranthus* some traditional uses in America, Asia, and Africa

## 8. Traditional Uses of *Amaranthus* in America

*Amaranthus* was cultivated 6000–8000 years ago by the mighty Aztecs. The Mixtecs of Oaxaca esteemed this plant. During the pre-Columbian civilizations, the turquoise mosaic covering the skull within Tomb 7 at Monte Alban was kept together by the sticky paste of *Amaranthus*. Though it was a staple food of the Aztecs, However, the cultivation of *Amaranthus* decreased and almost disappeared in Colonial times, under Spanish conquest [43,92]. In ancient times of the Mesoamerica, *Amaranthus* seeds were commonly used by the Aztec/Mexica. Hence, the cultivation of *Amaranthus* was significant because they use it for tribute payment [27]. It played a great role in their worship of gods [30,89]. During the festival called *Panquetzaliztli* (raising banners), the Aztecs used the mixture of *Amaranthus* and honey to build and bake images of their patron deity *Huitzilopochtli*. It is a ceremony where *Amaranthus* dough figurines of *Huitzilopochtli* were carried around in processions, after which. It is broken, distributed among the population for eating [93,94]. This practice made amaranths unable to survive as a staple food. All this happened on the arrival of the Spanish with Cortez when efforts were made to force Christianity on

the pagan natives, and *Amaranthus* grain was forbidden, fields burned, while cultivators were punished severely [30]. Currently, *Amaranthus* is regaining its importance. Hence, it is grown in a few locations in the United States [4]. More of the total species of *Amaranthus* exist in America, and they include: *A. cruentus A. hypochondriacus, A. caudatus, A. tricolor,* and *A. dubius, A. blitum, A. gangeticus, A. spinosus, A. viridis*. However, besides the common ones are, *A. retroflexus* and its congener, *A. hybridus is* native to North America, but is currently established as toxic weeds world-wide [24,95]. This implies that not all Amaranths are consumable, as some could be weed and perhaps toxic. Hunter and gatherers were the first to consume native food like *Amaranthus* in both North and South America before the domestication [96]. In America, *Amaranthus* seeds are consumed whole, it is toasted or milled into flour to make bread, noodles, pancakes, cereals, granola, cookies. In addition, the seed is popped like popcorn or flaked like oatmeal. Over 40 amaranth-based products are produced and are currently on the market in America. *Amaranthus* are also used for animal feed, it is used for textile dyeing, and it is among the ornamental plant; hence, it provides beauty to many environments. In the past two decades, there has been an increasing trend of the replacement of synthetic dyes with natural pigments from plants [14]. The pigment in *Amaranthus* can be used to prepare yellow and green dyes, red amaranth pigment has been used as a colorant in foods and medicines [14]. In America, the roots of *Amaranthus* have been used to produce dye though, it fades very slightly and resulted in a light color. In Bolivia and northwestern Argentina, people used red dye of *Amaranthus* leaves to color alcoholic beverages, while in Mexico and the Southwestern United States it is used for coloring maize dough and coloring foods and beverages [27].

Species of *A. cruentus* were primarily used by Mexicans to produce typical sweets called "alegera". The Alegera is a common traditional snack that is made by a process where Amaranth grains are toasted and mixed with honey or chocolate [27]. Also, *Amaranthus* grains are popped and mixed with honey to make *jaggery* or *molasses* as candy types of product [97,98]. It has been investigated that the green leafy *Amaranthus* possess the ability to maintain green foliage during the summer period. Therefore, it has been used as a rich source of complete amino acid, iron, and other essential nutrients for humans and livestock [24]. Although, ingestion of the fresh green or old *Amaranthus* leaves has been linked with out-breaks of livestock poisoning [95,99]. Before now, United State has been the leading producer of grain *Amaranthus*, and it is used in small scale businesses for making food products, but, in the last decade, China happens to be the largest grain *Amaranthus* producing area, though the Chinese uses *Amaranthus* as forage, rather than harvesting it as grain [27].

## 9. Uses of *Amaranthus* in Asia

In India, *Amaranthus* is a common wild vegetable. It is a weed that is cultivated and commonly known as "*Chowlai*." Notably, *Amaranthus* holds a vital role in Asian traditional (herbal) medicine. For example, *Amaranthus* spinous juice has been extracted and used by the tribe of Kerala to prevent swelling around the stomach, while the leaves are boiled alone and consumed for 2–3 days for curing jaundice [49]. Additionally, in Asia, the plant ash in a solution is used to wash sores, and the plant sap is used as an eyewash to treat ophthalmia as well as convulsions in children. In the Indian traditional system of medicine (Ayurveda), the plant is used as a febrifuge (medicine that reduces fever), laxative (a drug that stimulated bowel evacuation) effective diuretic (a drug that effectively stimulates passing out of urine) [49,100]. The Himalayans region consumed *Amaranthus* grain as a minor cereal. Moreover, its usage in food preparation, it is a popular medicinal plant used to treat bronchitis, appetizer, biliousness, galactagogue, *hematinic,* stomach effects, nausea, flatulence, anorexia, blood diseases, burning sensation, *leucorrhoea,* leprosy, piles, and as a treatment for hallucination, healing of wounds and rheumatism as well as arrest the coughing up of blood. All parts of the plant are known to contain medicinally active constituents [101,102]. An infusion of *Amaranthus* leaves is used by Sikkim when there is a stomach disorder, such as indigestion and anemia. Ethnic use of this plant is mainly with

village people of Sikkim [41,103]. Likewise, in India, the root extract is given as a vermicide among the Santhali and Paharia in eastern Bihar, while an aqueous decoction of the plant is given to check chronic diarrhea in Southern Orissa [41]. In addition, the Nepalese and some tribes in India apply *Amaranthus* spinous to induce abortion. Moreover, *Amaranthus* seeds are eaten raw in India and other places, it is mixed with other grains or processed into flour and is used to make food products. Some species have been used in desert areas of India, where the seeds of these species have been used as famine foods) [27]. Similarly, in South-East Asia, a decoction of the root is used to treat gonorrhea and is also applied as an emmenagogue (a substance that stimulates or increases menstrual flow) and antipyretic (an active medical substance that reduces fever) [103]. In mainland South-East Asia, the rainy season is linked with malaria-endemic. Hence, *Amaranthus* spinous bark decoction is taken in a volume of about one liter three times a day where malaria is prevalent. In India, the leaves of *Amaranthus caudatus* are used as a tea to relieve pulmonary problems and piles. They are used in the purification of blood, strangury in scrofula, and as a diuretic. The boiled leaves are used to trat swellings and stomach upset [13]. Other Asian countries have used *Amaranthus*: in traditional medicine, *Amaranthus* spinous is used to treat diarrhea. The root is also used for treating toothaches; in South east, in Malaysia, *Amaranthus* spinous is used as a cough syrup for relieving affected breathing in acute bronchitis [49]. The Chinese also use *Amaranthus* spinous in traditional medicine to treat diabetes. The seeds are used as a poultice for broken bones, smooth paste of leaves and roots have been explored as cataplasm, for a relief from skin diseases/disorders, such as abscesses, bruises, burn, eczema, inflammation, gonorrhea, menorrhagia and wound [49]. Still in Asia *Amaranthus* root paste with equivalent volume of honey has been used to control vomiting and, when mixed with sugar and water it helps in controlling dysentery. A mixture with black pepper in ration 1:3 that is (a part black pepper and 3 parts root paste) is administered twice daily, for tackling rabies [49].

## 10. Uses of *Amaranthus* in Africa

Although it is still unknown how *Amaranthus* arrived in Africa, however since 1980, *Amaranthus* has been identified as a promising food plant and a drought-resistant plant, a smart food for climate change [4]. It is widely found in Africa as an ornamental plant or weed throughout the continent, from Senegal to Nigeria in West Africa, and from Equatorial Guinea to Zaire central Africa [4]. Species have also been identified in Morocco, Ethiopia, and Sudan [4]. African traditional vegetables and *Amaranthus* are progressively recognized as promising food plants with both macro-nutrients and micronutrients, besides its being a source of bioactive compounds in the diets of most rural populations in Africa [4]. The continent is endowed with varieties of other vegetables. The most common African leafy vegetable is *Amaranthus*. Although it is agreed that about 60–75 species of *Amaranthus* exist, only 17 of the species are edible leaves and three-grain purposes species [4]. Several species are often considered weeds because they grow naturally on fallow lands and even by the roadsides with no attention [10]. A few species among the leafy species are grown as the most common species in tropical Africa markets. Grain amaranth is not commonly cultivated in Africa. Many of the rural populations don't utilized the seed as food because they are less informed that Amaranthus seeds are edible, a such, where Amaranths are grown for seed purposes, they must be the introduced varieties of American origin [104]. In recent times, however, few farmers have been engaged seriously in the growing of grain *Amaranthus* such that they are supplying some millers and supermarkets in Zimbabwe, Kenya, Uganda, and Ethiopia [4,31]. Across Africa, most of the staple foods are predominantly starchy since they include cereals, such as maize, rice, wheat, and root crops. Most staple foods are considered inadequate in essential amino acids, such as lysine and tryptophan [105]. Traditional vegetables, such as *Amaranthus*, are suggested to complement the nutritional inadequacies of such starchy staples [24]. Phytochemicals and bioactive compound are present in *Amaranthus* leave hence are also recognized as anticancer potential, which benefits humans, by inhibiting the proliferation of liver, breast,

and colon cancer cell [106,107]. There are still gaps in the information on the potentials and uses of the popular *Amaranthus* in Africa. *A. cruentus* is common in Africa, and *Amaranthus caudatus* are widely used as ornamental and grain *Amaranthus*. The variety of *cruentus* was introduced recently to Southern Dahomey (Benin), which proved to be drought resistant. The third American *Amaranthus* (*A. hypochondriacus*) was introduced into East Africa in 1940 [4]. In Africa, traditional vegetables like *Amaranthus* appear to be the most recognized, perhaps. It is the most affordable among other traditional vegetables [4]. It is high in fiber, low in calorie, hence, is used to supplement cereal-based food, such as maize and wheat [108]. Hence, leafy vegetables are increasingly recognized as promising contributors to human nutritional needs, which are capable of supply both macro and micro-nutrients besides their bioactive compound benefits [106].

The research focus on *Amaranthus* spp. has been rapidly expanding, and many reports have been published, especially on the grain's amaranth [97,109]. Several studies focused on different aspects, such as botanical, agrotechnological, biological, chemical, and technological, and nutritional composition properties [110]. In Africa, the cultivation and consumption of indigenous leafy vegetables such as *Amaranthus* species are highly dependent on factors such as poverty status or degree of urbanization. This is because poor families tend to consume cheap leafy vegetables more than wealthy families; hence, Amaranth is considered as the poor man's vegetable since it is one of the less expensive leafy vegetables in the markets [4]. In Sub-Saharan Africa, important nutrients are lacking in the staple diets of most of the population because their diet mainly consists of maize, sorghum, rice, and cassava. These crops do not take up zinc while growing, and therefore, cannot provide it in the diet. Hence they must be supplemented [4,27]. Furthermore, several African countries have used *Amaranthus* as an important dietary supplement for those infected with HIV/AIDS.

Thirteen species of *Amaranthus* were found to have been documented as commonly found in the Western part of Africa, precisely in Nigeria under cultivation, either as weeds or ornamental. Although none of the thirteen species is of Nigerian origin, most have since adapted to the climate and are valued for their leaves, herbaceous stem, inflorescence, seeds, and chemical byproducts [51]. In West Africa, Nigeria, Amaranths are mostly considered vegetables with great potentials. The leaves, the stems are utilized in potherb, salads, burgers, and soup or stew. *Amaranthus* is cheap; hence it has been a preferred choice for many. It is also preferred because it can supplement the diverse cereals and legumes, which make up the bulk of daily food [51]. The most common mode of utilizing *Amaranthus* species in Africa, for example, in Nigeria, is to cook the leaves like spinach. The leaves are also added toa mixture of salads [51]. West Africans value the leafy amaranths more than the grain species, perhaps because of the limited information on whether the seeds are edible not. However, the leaves are collected during the growth cycle when the nutrient is at its peak and are mixed with condiments to make soup. At the same time, the stems are also preferred fresh and succulent. The grain amaranths are underutilized in Nigeria despite their potential to contribute to food security and livelihood. They are also used as ornamentals to decorate parks, domestic and office complexes [51].

In South Africa, poor households tend to use traditional leafy vegetables, especially *Amaranthus*, more than their wealthier counterparts. This is simply because traditional vegetables such as *Amaranthus* are cheaper in terms of money, accessibility, and even cultivation than exotic vegetables, which are expensive [11,51]. Therefore, using wild food, such as *Amaranthus*, forms part of the safety measure that rural dwellers use as a coping mechanism to mitigate poverty, disaster, and livelihood stresses. Similarly, consumption of leafy vegetables is highly dependent on factors, such as poverty status, degree of urbanization, accessibility to fresh produce markets, and season of the year. In South Africa, the young leaves of *Amaranthus* are cooked and used as vegetables [11]. Similarly, in the past, in Limpopo Province, South Africa, where there was difficulty in accessing salt, dried *Amaranthus* plants of different species have been burnt to produce ash, it is dissolved in water, and the filtrate of the ash was used as salt [11]. Regionally, in Southern Africa,

*Amaranthus* leaves are harvested by hand and are best eaten fresh, just as spinach. It is boiled in salted water and served with a sauce of tomatoes and onion or potatoes. The fresh leaves are added to salads or stir-fry and added to other cooked vegetables, such as broccoli and onions. In addition, a cup of boiling water could be poured on a quarter cup of fresh *Amaranthus*, and the infusion of *Amaranthus* leaves used to treat anemia, chronic fatigue, diarrhea as well as coughs, and heavy menstrual bleeding. Amaranth also has got other uses: in the Tzaneen area, *Amaranthus spinous* leaves and stems are dried, grounded, and are used as snuff [111]. In addition, a cooled tea of *Amaranthus* can be used as a lotion to relieve itchy burning skin and is used to clean wounds [112].

Most Amaranth species consumed appeared to be of the wild species. Very few are grown, which fall within leafy types that are most common in markets of tropical Africa. Although *Amaranthus* also has been grown for their seed's purposes, in the continent, however, they include those varieties of American origin, and many do not utilize their seeds [113]. There is an increased production and consumption of leafy vegetables in Rwanda, Uganda, Malawi, and Tanzania [4]. Another important project was the Diversity International's African leafy vegetable program conducted in Botswana, Cameroon, Kenya, Senegal, and Zimbabwe, which induced notable changes in growing, consumption, marketing, and nutritional awareness of African leafy vegetables, including Amaranths [114]. There are still gaps in the knowledge of some popular *Amaranthus* in Africa, resulting in complexities and confusion in the species' nomenclature and their nutritional composition [104]. Moreover, there is limited information on the breeding potentials, especially of the wild species, that can be promoted for sustainable utilization, especially among the rural communities [4]. Interventions have been carried out by partnerships between research institutions and local NGOs to promote indigenous vegetables, including *Amaranthus*. A typical example of a project led by the World Vegetable Center was centered on the promotion of neglected indigenous vegetable crops for improved nutrition and health in Eastern and Southern Africa [4]. The leafy *Amaranthus* is the most commercialized in tropical Africa [4,113]. The acceptability, utilization and commercialization of Amaranths could be improved since research are on the rise, creating more awareness that could promote its usefulness, accessible strategy, and utilization [4].

## 11. Problems Associated with Production and Underutilization of *Amaranthus*

Many underutilized vegetables, including *Amaranthus* surfers, neglect as a result of a preference for exotic vegetables, which affects *Amaranthus* production and its utilization [115]. Also, there is an increased transition of the dietary lifestyle of people to fast food, which affects the utilization of traditional vegetables because it appears the younger generation prefers most fatty, sugary, salty tastes associated with many snacks and fast foods [116]. The methods of preparation and other complementary ingredients used during cooking might also be a problem for consumers. Hence, a more novel method of preparation of *Amaranthus*-based food products is suggested. In the same vein, the negligence could also be attributed to people's perceptions and preferences [115,116]. The loss of the traditional food knowledge system due to social changes have been recorded in Africa as a factor for the under-utilization of most traditional vegetables, including *Amaranthus* [115]. People's attitudes and perceptions are identified as contributing factors to the neglect of many traditional vegetables, including the wild varieties of *Amaranthus,* which has been considered as pig feeds; hence it is under-utilized. However, *Amaranthus* is reported as an edible plant and a valuable source of nutrients for the optimum wellbeing of both humans and animals aside from other uses [27]. In addition, when *Amaranthus* is grown on nitrogen-rich soils, which may concentrate nitrate in the leaves or any tissue of the plant. Nitrates may contribute to stomach cancers, blue babies, and other health problems [117]. This is because *Amaranthus* is a plant that has a high chance to accumulate nitrates, especially when soil fertility is very high. Worst is when nitrogen fertilizer is added, and if things like herbicide, drought, or frost slow the processes of photosynthesis, Nitrates can only concentrate mainly in the plant tissues and not in the seeds [117]. Interestingly,

humans and some animals have bacteria in their digestive systems that convert nitrates to nitrite; such that by six months of old in humans, the acid levels in the digestive system rise higher; hence it kills these bacteria, and the danger of nitrate poisoning is unlikely, to occur. However, there is an increased risk if a pregnant woman with low stomach acidity is being treated for cancer [117]. Therefore, it advisable to eat this plant if it is grown on natural land where chemical fertilizers are not used. Poor cultivation practices of plant foods may predispose the consumer to some health risks of nitrates and oxalates and possible contamination with toxic materials like lead, resulting from the poor cultivation practices or pollution, which can contribute to its underutilization [118].

## 12. Nutraceutical and Healing Potentials of *Amaranthus*

Nutraceutical's properties are the non-specific biological agents that boost and promote well-being. It can serve as a preventive measure to malignant processes and controls the symptoms of illnesses [119]. The wild species of *Amaranthus viridis* and *spinosus* have been investigated for their composition, including their medicinal and nutraceuticals properties. The Weedy *Amaranthus* has been identified to have remarkable protein, dietary fiber, carbohydrates, Fe, Ca, Zn, K, Mg, P, S, Mn, Cu, Na, chlorophylls, β-cyanins, β-xanthins, betalains, β-carotene, vitamin C, compared to other cultivated species.

Both seeds and *Amaranthus* leaves are highly nutritious for human consumption. It promising to meet the nutritional need hence can be explored as a food plant with an added advantage that can either be explored for its preventive or curative purposes. Several studies report that *Amaranthus* seed or oil may benefit those with hypertension and cardiovascular disease; hence, its regular consumption reduces blood pressure and cholesterol levels while improving the nutritional value of essential micro-nutrients, including β-carotene, iron, calcium, vitamin C, and folic acid [4,14,120]. The leaves have great potential for pro-vitamin A carotenoid (β-carotene), which are boosters of immunity. For example, the amino acid profile of *Amaranthus cruentus* leaves includes appreciable methionine niacin and lysine levels, which are the limiting amino acids in most plant proteins [4]. Also, the seed, leaves, stem, and root carotenoid content of *Amaranthus cruentus* was investigated, and the leaves were found to be highest in β-carotene [48]. Currently, it appears *Amaranthus* is advancing to be a new millennium crop with nutraceutical value. The major carotenoid identified in *Amaranthus* leaves canthaxanthin, which is an anticancer agent, other carotenoid includes β-carotene and lutein, an active nutrient essential for the prevention of age-related eye diseases [24,52]. *Amaranthus* is rediscovered as a promising plant with the potential to provide high-quality protein, unsaturated oil [52]. *Amaranthus* leaves are a great source of proteins and micro-nutrients, including iron, calcium, zinc, vitamin C, and vitamin A and B vitamins. Even though studies on the leaves of *Amaranthus* are limited, in Africa, leafy *Amaranthus* vegetables provide many rural households with food, especially in the summer and winter seasons [4]. There is usually sudden rise of food prices during the winter season, hence, most households fall back to traditional vegetables, such as *Amaranthus*. Hence, it forms the basis of nutrition in most rural households, and it is contributing up to about 80% of their total food supply, excluding the maize component of food used in winter. This dilemma is evident among communities with high unemployment issues, especially where children no longer qualify for social grants like child support grants. Thus, low-income earners tend to be more reliant on traditional vegetables. This substantiates a link between *Amaranthus* and poverty, as reported in recent scientific studies [4,121].

Several studies have shown that oil is being extracted from *Amaranthus*, both the seed or oil can benefit those with cardiovascular disease, including hypertension, the regular consumption of *Amaranthus* reduces blood pressure, cholesterol levels, improves the antioxidant status and immune parameters that optimum wellbeing. In Benin, burnt dried *Amaranthus* plants are used to make potash. Vegetable *Amaranthus* has been used as a good source of medicinal properties that can be beneficial to young children, lactating mothers, and other patients with constipation, also fever, hemorrhage, anemia, or kidney

complaints [4,27]. In Africa, an ethanolic formulation of *Amaranthus spinosus* increased hemoglobin content and blood schizonticides activity, which is reported to use as an antimalarial effect compared to chloroquine [24]. The paste of leaves and roots of the same species have been applied as a poultice to relieve skin diseases/disorders that include: abscesses, bruises, burn, eczema, inflammation, gonorrhea, menorrhagia, and wound [49].

*Amaranthus* has soporific effects and febrifuge actions, which are also used to induce sweat and to reduce fever. The boiled leaves are administered for 2–3 days to cure jaundice, some kinds of rheumatic pain, and stomach ache [24,122]. In addition, the paste of the root has been used to possess several beneficial effects when used internally and externally [49]. The root paste with an equal volume of honey controls vomiting when mixed with sugar, and water controls dysentery (stools and mucus), when mixed with black pepper in ratio 1:3 proportion meaning (one part black pepper and three parts root paste) can be administered twice daily is beneficial in Rabies [49]. The entire plant of *Amaranthus spinosus* has been used as diuretic, purgative, refringent, and to treat cholera, piles, and snake bit [49,60]. A study in Kenyan, the South Coast in the Msambweni community, to be precise, where malaria is endemic, reports that *Amaranthus hybridus* L, a plant species, has been used traditionally to treat malaria [24].

The roots of *Amaranthus* can be boiled with honey and used as a laxative for infants. In Ghana, the water of macerated *Amaranthus* plants has been used as a wash to treat pains in the limbs. Similarly, in Ethiopia, *A. cruentus* is used as a tapeworm excellent. Moreover, in Sudan, the ash from the stems is used for wound dressing. The heated leaves are used in the treatment of tumors. Furthermore, *A. tricolor* and *A. caudatus* are used externally to treat inflammations and internally as a diuretic diet [123]. Moreover, the seeds of *A. spinous* are used as a poultice for broken bones. It is used internally to treat internal bleeding, diarrhea, and excessive menstruation [47,49]. In Southeast Asia, a decoction of Amaranthus root is used to treat gonorrhea. It is also applied as an emmenagogue and antipyretic [4,91]. Among the Nepalese and some tribes in India, *Amaranthus spinous* is used to induce abortion [24]. Countries across the world considered the bruised leaves as a good emollient [123]. Likewise, the leaves are used for gastroenteritis, gall bladder inflammation, abscesses, arthritis and are used to treat snakebites. In addition, the sap of *Amaranthus* has been used as an eyewash to treat ophthalmia and convulsions in children. Hence, these reports attest to a rise in interest in medicinal plants that have healing effects on humans. *Amaranthus* is a source of various phenolic phytochemicals, including rutin, isoquercitrin and nicotiflorins, to mention a few. It has tremendously been used in people's daily food and industries. In addition, these phytochemicals are used in various medical fields either as preventive and curatives to diseases [106]. Interestingly, *Amaranthus* is enlisted among the medicinal plants, which have been explored to possess great potentials for treating several illnesses. Furthermore, extracts from *Amaranthus* has been used for medicinal purpose in Asia and ancient Indian, Nepalese, Chinese, and Thai to treat several health conditions including urinary infections, gynecological conditions, diarrhea, pain, respiratory disorders, diabetes, and as a drug that increased the passing of urine [24]. The root extract of *Amaranthus spinous* has been used as vermicide (a poisonous substance to worms), whereas, the aqueous decoction/infusion of the plant is used for chronic diarrhea in southern Odisha [24]. Some tribes apply spiny *Amaranthus* to induce abortion. Still, on *Amaranthus*, the juice is used by tribal of Kerala, a state in India to prevent swelling around the stomach, while leaves are boiled without salt and consumed within 2–3 days to cure jaundice [124]. However, the anticancer, anti-viral, hepatoprotective, neuroprotective, cardioprotective and antidiabetic properties of *Amaranthus* with relevance to current global health scenario are currently in the public interest [4]. Scientific interest in *Amaranthus* and its health promoting benefits has increased significantly in the recent past with various analyses presenting nutraceutical properties of amaranth [124,125].

It is established in several studies that all parts of *Amaranthus* spp. are edible [4,14]. Holistically, the plant *Amaranthus* is a rich source of nutraceutical, which are yet to be exploited to their maximum benefits. Although the leaves of *Amaranthus* are usually

neglected in research, an example of a study on *Amaranthus cruentus* was investigated for its carotenoid content in Dalmatia and, the report of pro-vitamin A carotenoid (β-carotene) content was interestingly revealed in the following order: Carotenoid content was identified highest in the leaves, followed by seeds, stem, and roots [126]. The major carotenoid identified in the leaves was canthaxanthin, an antitumor agent, followed by β-carotene and lutein, which is also considered active in retarding age-related eye diseases [24,126]. The β-carotene of *A. cruentus* in that study was seven times higher than in tomatoes, which may help to treat anemia in African countries [24,126].

In addition, *A. spinosus* have been reported to have higher calcium content in the dry leaves, followed by *A. tricolor*, *A. viridis*, and *A. blitum*, while iron content was rated highest in *A. viridis* followed by *A. spinosus*, *A. tricolor*, and *A. blitum*. This implies that *Amaranthus* spp. can be explored as a source of biogenic calcium and in antacid preparations [43].

Amaranth grain, commonly known as *Rajgira*, in India, has been identified as highly nutritious and a good source of bioactive compounds such as anthocyanins and polyphenolics. Furthermore, the grain has been noted for its high protein content, with complete essential amino acids and substantial lysine as unique compared to other grains. Besides, it is known for quality starch, oil, fiber, vitamins (A, K, B6, C, E, and B), and essential minerals like calcium, iron, etc. [24] It also have advantages that can be explored as a food supplement since it is gluten-free with a high content of quality protein and unsaturated fatty acids. Grain *Amaranthus* is a rich source of fiber and an alternative natural source of squalene (a triterpene), a superior antioxidant ever identified for its wide biological efficacy; against cancer [127]. Hence, Antioxidants, molecules with the reducing effect of free radicals vital for protection against cancer and degenerative disorders, are in abundant *Amaranthus* spp. [24]. The antioxidant potential of *Amaranthus* has been credited to the presence of appreciable levels of phenolics and flavonoids. The leaves and flowers of *Amaranthus* and their extracts have been investigated to possess the highest antioxidant activities compared to other dietary rutin supplements [24]. *Amaranthus* is identified to contains nutraceutical elements when compared with the exotic vegetables. An example of Amaranthus nutritional value compared with an exotic vegetable is described in Table 4, where it has been identified to have an appreciable amount of nutrients, hence it can be included in the diet or as a daily nutritional supplement for addressing malnutrition diseases. The inclusion of *Amaranthus* in diet especially the staple foods, can boost immunity and optimize well-being

**Table 4.** Nutritional value of raw and cooked (boiled and drained without salt) amaranth leaves compared with raw cabbage. Source [3].

| Nutrient/Leafy Vegetable | Raw | Cooked | Cabbage |
|---|---|---|---|
| Protein | 1.2 | 2.11 | 1.28 |
| Calcium | 40 | 209 | 40 |
| Iron | 0.47 | 2.26 | 0.47 |
| Magnesium | 12 | 55 | 12 |
| Phosphorus | 26 | 72 | 26 |
| Potassium | 170 | 641 | 170 |
| Manganese | 0.160 | 0.861 | 160 |
| Vitamin C | 36.6 | 41.1 | 36.6 |
| Riboflavin | 0.04 | 134 | 0.04 |
| Niacin | 0.234 | 0.658 | 0.559 |
| Vitamin A | 5 | 139 | 5 |

## 13. Conclusions

*Amaranthus* has been identified as an underutilized vegetable with great potential to supply the dietary needs of animal and humans. Although Amaranthus is low in calories, its protein, fat, and fiber, as well as essential vitamins and minerals, are considered to alleviate malnutrition and optimize well-being. *Amaranthus* can play a significant role in the

food and nutrition security of vulnerable groups in urban and rural settings. *Amaranthus* has been rediscovered to have medicinal, nutraceutical, industrial, and ornamental benefits, which cannot be overemphasized. Currently, *Amaranthus* is considered as one of the most produced and consumed indigenous vegetables on the African continent with high nutritional potentials, which are yet to be explored. The potential and use of traditional vegetables (*Amaranthus*) varieties in America, Asia, and Africa has been reviewed and, the cultivation of *Amaranthus*, commercialization and processing of the leaves to powder is still at a low level. Suitable production systems, innovative processing, and novel value-adding techniques that may promote utilization of *Amaranthus* are lacking. Similarly, education about Amaranthus nutritional benefits and the provision of good packaging and storage is needed. The addition of *Amaranthus* as an ingredient in indigenous foods could enhance the utilization of *Amaranthus* at the community level which can improve the food and nutrition security of population with nutrient deficient challenges and could also make *Amaranthus* a viable commercial food product for improved livelihood of many, especially the under-privileged.

**Author Contributions:** The review for this publication was conceptualized by O.N.R., under the guidance of M.C., N.N. and K.U. O.N.R. designed and wrote the manuscript under the guidance and supervision of the N.N. and K.U. All authors have read and agreed to the published version of the manuscript.

**Funding:** This review receives no funding.

**Institutional Review Board Statement:** Not applicable.

**Informed Consent Statement:** Not applicable.

**Data Availability Statement:** The images described in Figures 1 and 2 was the only data obtained from my field study, otherwise, this review article did not report any data. Peer-reviewed articles online, and book materials was used for the data in this review.

**Conflicts of Interest:** The authors declare no conflict of interest.

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
