# Peer review of "Underutilization Versus Nutritional-Nutraceutical Potential of the Amaranthus Food Plant: A Mini-Review"

_applsci, doi:10.3390/app11156879_

Round 1

Reviewer 1 Report

The review of Ruth et al. reports on the proprieties, the uses, the underutilization of Amarantus all around the world. The paper is interesting but a bit long in some points. For example, too many details are given for the traditional uses of Amaranthus in America, Asia and Africa. In my opinion this part could be shortened.

It could be useful to move the figures before in the text.

The paper seems more appropriate for a journal as Plants or Sustainability.

The title must be corrected.

Some highlighted text is found to be cancelled.

Author Response

Reviewer 1

The review of Ruth et al. reports on the proprieties, the uses, the underutilization of Amaranthus all around the world. The paper is interesting but a bit long in some points. For example, too many details are given for the traditional uses of Amaranthus in America, Asia, and Africa. In my opinion, this part could be shortened.

The details in the traditional uses of Amaranthus in America, Asia, and Africa have been revised and shortend.

It could be useful to move the figures before in the text.

The document has been formatted according to the journal’s requirement.

The paper seems more appropriate for a journal as Plants or Sustainability.

The manuscript’s focus is on nutritional-nutraceutical endowment, and the food and nutrition security potential of Amaranthus plant and its underutilization, which is in line with the Special Issue ("Underutilized Natural Sources in Food and Dietary Supplements") under Applied Science Journal.

The title must be corrected.

The title has been corrected and now reads:

“Underutilization versus Nutritional-nutraceutical potential of the Amaranthus food plant: A mini review”

Some highlighted text is found to be canceled.

The highlighted text has been canceled.

Reviewer 2 Report

The paper is a nice mini review on the past and future perspectives for the genus Amaranthus. It has promising potentialities, but I believe in its current status it severely lacks a robust and rational structure.

These are the major points authors should look at in my humble opinion:

  1. English editing - it is in a few paragraphs (ie abstract) not acceptable.
  2. Structure. Authors should clearly differentiate among plant parts and typologies of uses. In my opinion they should built the following sections: A. ethnolinguistic of the Genus Amaranth: folk names in the world and their meaning, possibly dividing them by continents (ie. pigweed was the name in English simply because the plant was used to feed the pig and SURELY THIS IS NOT THE REASON WHY THE PLANT IS UNDERESTIMATED AS ERRONEOUSLY WRITTEN IN THE ABSTRACT); B. food uses (separating seeds/fruits and leaves) divided by geographical macro-areas; C. medicinal uses divided by geographical macro-areas; D. veterinary uses (both as fodder and medicine for animals), divided by geographical macro-areas; E. other uses, divided by geographical macro-areas; F: perspective for the future
  3. Some very crucial literature is fully missing: Amaranth is a CRUCIAL wild vegetable in some area of the Mediterraean, see esp. in Southern Italy (check for example works by Geraci, Tuttolomondo, Lentini, Pieroni, Nebel, Guarrera), to less extent in the Iberian Peninsula (check works by Valles, Pardo, and Tardio research groups), and, last but not least, especially in the Balkans including Greece - Amaranthus is  the most important Romanian/Vlach wild food veggie! - Pls check the ethnobotanical works by Hajdari, Dogan, Mustafa, Nedelcheva, Pieroni, and others.
  4. In the last section (F. Perspective for the future) the authors should clearly spell out HOW Amaranth uses could be promoted at the community-level.

Author Response

Reviewer 2

The paper is a nice mini-review on the past and future perspectives for the genus Amaranthus. It has promising potentialities, but I believe in its status it severely lacks a robust and rational structure.

The robust and rational structure of the review has been improved on.

These are the major points authors should look at in my humble opinion:

English editing - it is in a few paragraphs (i.e., abstract) not acceptable.

The abstract has been proofread for English language errors accordingly.

Structure. Authors should clearly differentiate among plant parts and typologies of uses. In my opinion they should built the following sections: A. ethnolinguistic of the Genus Amaranth: folk names in the world and their meaning, possibly dividing them by continents (ie. pigweed was the name in English simply because the plant was used to feed the pig and SURELY THIS IS NOT THE REASON WHY THE PLANT IS UNDERESTIMATED AS ERRONEOUSLY WRITTEN IN THE ABSTRACT); B. food uses (separating seeds/fruits and leaves) divided by geographical macro-areas; C. medicinal uses divided by geographical macro-areas; D. veterinary uses (both as fodder and medicine for animals), divided by geographical macro-areas; E. other uses, divided by geographical macro-areas; F: perspective for the future.

In line with the scope and focus of the paper, the manuscript has been structure accordingly.

“pigweed” as name for Amaranthus is probably a local name or local parlance for Amaranthus those region.

Some very crucial literature is fully missing: Amaranth is a CRUCIAL wild vegetable in some area of the Mediterraean, see esp. in Southern Italy (check for example works by Geraci, Tuttolomondo, Lentini, Pieroni, Nebel, Guarrera), to less extent in the Iberian Peninsula (check works by Valles, Pardo, and Tardio research groups), and, last but not least, especially in the Balkans including Greece - Amaranthus is the most important Romanian/Vlach wild food veggie! - Pls check the ethnobotanical works by Hajdari, Dogan, Mustafa, Nedelcheva, Pieroni, and others.

This review is a mini review with focus on raising awareness on food and nutrition security potential of Amaranthus plant, hence, some literature were excepted to maintain the core interest of the review.

In the last section (F. Perspective for the future) the authors should clearly spell out HOW Amaranth uses could be promoted at the community-level.

This has been revised and stated accordingly in the concluding section.

Summarily, Amaranthus uses could be promoted at the community-level by various means which include:

  1. Organizing sensitization workshops as well as seminars on Amaranthus especially in local languages and local parlance; creating awareness regarding the use of Amaranthus as a food plant and supplement, highlighting its nutritional potentials and benefits.
  2. Good packaging and proper storage to promote its acceptability and commercial viability in local communities.
